# Head-to-Head Comparison of Neck ^18^F-FDG PET/MR and PET/CT in the Diagnosis of Differentiated Thyroid Carcinoma Patients after Comprehensive Treatment

**DOI:** 10.3390/cancers13143436

**Published:** 2021-07-09

**Authors:** Yangmeihui Song, Fang Liu, Weiwei Ruan, Fan Hu, Muhsin H. Younis, Zairong Gao, Jie Ming, Tao Huang, Weibo Cai, Xiaoli Lan

**Affiliations:** 1Department of Nuclear Medicine, Union Hospital, Tongji Medical College, Huazhong University of Science and Technology, Wuhan 430022, China; songymh@hust.edu.cn (Y.S.); fangliu72@hust.edu.cn (F.L.); 2017xh0388@hust.edu.cn (W.R.); hufan1201@163.com (F.H.); gaobonn@hust.edu.cn (Z.G.); 2Hubei Province Key Laboratory of Molecular Imaging, Wuhan 430030, China; 3Department of Radiology, University of Wisconsin-Madison, Madison, WI 53705, USA; myounis@wisc.edu (M.H.Y.); wcai@uwhealth.org (W.C.); 4Department of Medical Physics, University of Wisconsin-Madison, Madison, WI 53705, USA; 5Department of Breast and Thyroid Surgery, Union Hospital, Tongji Medical College, Huazhong University of Science and Technology, Wuhan 430022, China; mingjiewh@hust.edu.cn (J.M.); huangtaowh@163.com (T.H.)

**Keywords:** thyroid neoplasms, fluorodeoxyglucose F18 (^18^F-FDG), positron emission tomography (PET), magnetic resonance imaging (MRI), recurrence, neoplasm metastasis

## Abstract

**Simple Summary:**

The most advanced positron emission tomography–magnetic resonance (PET/MR) combines the high soft tissue contrast of MRI with the high functional/metabolic sensitivity of PET and has the potential to achieve the highest level of diagnostic performance for refractory malignancies in differentiated thyroid cancer (DTC) patients. The utility of PET/MR in the postoperative follow-up of DTC patients has been relatively ambiguous. This retrospective study compared ^18^F-fluorodeoxyglucose neck PET/MR with PET/CT head-to-head, in order to evaluate the diagnostic efficacy of PET/MR in assessment malignancy in DTC patients after comprehensive treatment. We determined that PET/MR presented better detection rates, image conspicuity, and sensitivity than PET/CT in recurrent DTC lesions and cervical lymph node metastases. The addition of neck PET/MR scan after whole-body PET/CT may provide more favorable diagnostic information.

**Abstract:**

We explored the clinical value of ^18^F-FDG PET/MR in a head-to-head comparison with PET/CT in loco-regional recurrent and metastatic cervical lymph nodes of differentiated thyroid carcinoma (DTC) patients after comprehensive treatment. ^18^F-FDG PET/CT and neck PET/MR scans that were performed in DTC patients with suspected recurrence or cervical lymph node metastasis after comprehensive treatment were retrospectively analyzed. Detection rates, diagnostic efficacy, image conspicuity, and measured parameters were compared between ^18^F-FDG PET/CT and PET/MR. The gold standard was histopathological diagnosis or clinical and imaging follow-up results for more than 6 months. Among the 37 patients enrolled, no suspicious signs of tumor were found in 10 patients, 24 patients had lymph node metastasis, and 3 patients had both recurrence and lymph node metastases. A total of 130 lesions were analyzed, including 3 malignant and 6 benign thyroid nodules, as well as 74 malignant and 47 benign cervical lymph nodes. Compared with PET/CT, PET/MR presented better detection rates (91.5% vs. 80.8%), image conspicuity (2.74 ± 0.60 vs. 1.9 ± 0.50, *p* < 0.001, especially in complex level II), and sensitivity (80.5% vs. 61.0%). SUVmax differed in benign and malignant lymph nodes in both imaging modalities (*p* < 0.05). For the same lesion, the SUVmax, SUVmean, and diameters measured by PET/MR and PET/CT were consistent and had significant correlation. In conclusion, compared with ^18^F-FDG PET/CT, PET/MR was more accurate in determining recurrent and metastatic lesions, both from a patient-based and from a lesion-based perspective. Adding local PET/MR after whole-body PET/CT may be recommended to provide more precise diagnostic information and scope of surgical resection without additional ionizing radiation. Further scaling-up prospective studies and economic benefit analysis are expected.

## 1. Introduction

Thyroid carcinoma is the most common endocrine malignant tumor worldwide, accounting for 2% of all cancers, after a two-fold increase over the last 25 years [1,2,3]. More than 90% of thyroid carcinomas are differentiated thyroid carcinoma (DTC), which includes papillary carcinoma (PTC) (85%) and follicular carcinoma (FTC) (12%). The prognosis of DTC is generally favorable after comprehensive treatment including surgery, radioactive iodine, and thyroid-stimulating hormone (TSH) suppression [4]. Nevertheless, up to 30% of patients may experience local recurrence and/or metastasis within several decades, which indicates a poor prognosis and a drop of the five-year survival rate from higher than 90% to 35–85% [5,6]. Postoperative recurrence appears most frequently (60–75%) in cervical lymph nodes (LNs) [7]. As a result, strict postoperative follow-up and advances in early detection are essential for a timely intervention in case of relapse and metastatic disease. Conventional ^131^I whole-body scan (^131^I-WBS), in association with periodic evaluation of serum thyroglobulin (Tg) and neck ultrasound, have been employed as the routine diagnostic procedure in the protocol of patients thyroidectomized for DTC [8,9]. However, 10–15% of follow-up DTC patients appear with abnormal thyroglobulin levels and negative findings on ^131^I-WBS [10,11,12]. In guidelines of different countries, the clinical indication of ^18^F-fluorodeoxyglucose positron emission tomography–computed tomography (^18^F-FDG PET/CT) has been widely accepted for postoperative DTC patients who present with the aforementioned discordant findings, as well as for the systemic assessment of patients with suspected metastases [8,9,10,13,14,15]. Nevertheless, according to previous reports, the sensitivity and specificity of ^18^F-FDG PET/CT in detecting DTC recurrence or metastasis are 46–100% and 66–100% respectively, which is considered inadequate [16,17].

Magnetic resonance imaging (MRI) is a useful technique for the diagnosis of thyroid nodules and metastatic cervical lymph nodes, which benefits from excellent soft-tissue contrast, superior spatial resolution, and the ability to functionally characterize tissues by utilizing non-contrast- or contrast-enhanced techniques such as diffusion-weighted imaging (DWI) and apparent diffusion coefficient (ADC) [18,19,20]. The hybrid PET/MR is a promising imaging modality that combines the high soft tissue contrast of MRI with the high functional/metabolic sensitivity of PET without additional ionizing radiation. For head and neck oncologic imaging, it has the potential to achieve the highest level of diagnostic performance. However, to date, the usefulness of PET/MR in head and neck malignancy has not been fully elucidated [21,22,23]. It is questionable whether simultaneous PET/MR can provide better diagnostic ability than CT, MRI, and PET/CT in loco-regional recurrent and metastatic cervical lymph nodes [16,24,25,26].

The purpose of the present study was therefore to evaluate the diagnostic ability of simultaneous neck PET/MRI in a head-to-head comparison with PET/CT for the assessment of malignancy in postoperative differentiated thyroid carcinoma (DTC) patients.

## 2. Materials and Methods

### 2.1. Patients and Lesions

We retrospectively reviewed the images of DTC patients who underwent ^18^F-FDG PET in our center from 23 October 2017 to 29 July 2020. The inclusion criteria were as follows: (1) diagnosis of DTC was confirmed by histopathological analysis; (2) patients had received comprehensive treatment (including total or subtotal thyroidectomy, ^131^I ablation therapy, or/and thyroid hormone replacement/suppression therapy); (3) patients had one or more of the following high-risk features of recurrence or cervical lymph node metastasis: (1) positive serum Tg and negative ^131^I-WBS, (2) rising anti-thyroglobulin antibodies (TgAb) after radioactive iodine ablation, (3) suspected widespread metastases throughout the body, eligible for ^18^F-FDG PET examination according to the available guidelines at that time; (4) patients accepted ^18^F-FDG PET/CT and subsequent PET/MR examination; (5) patients were available for follow-up including postoperative pathology/fine-needle aspiration biopsy (FNAB) or regular ultrasonography and Tg/TgAb level monitor every 3–6 months. This retrospective study was approved by the Ethics Committee of Union Hospital, Tongji Medical College, Huazhong University of Science and Technology. All patients signed an informed consent before undergoing ^18^F-FDG PET/MR imaging.

The local thyroid region lesions and cervical lymph nodes included in the present study were involved according to the following criteria: (1) the lesion was diagnosed by pathology after reoperation or FNAB; (2) the lesion was monitored regularly by imaging including neck ultrasonography, ^131^I-WBS, CT, and/or MRI, for at least 6 months.

### 2.2. ^18^F-FDG PET/CT Scan

^18^F-FDG was produced by a GE Medical Cyclotron (Mini trace, GE Healthcare, Milwaukee, WI, USA) and synthesized by the Tracelab MX FDG (GE Healthcare, Milwaukee, WI, USA) automatic synthesis system. The radiochemical purity was more than 95%. All patients were fasting for more than 6 h, with fasting blood glucose ≤ 200 mg/dL before injection. The patients were given an intravenous injection of ^18^F-FDG 3.70–5.55 MBq/kg according to their body weight. After resting for about 60 min, drinking 200–500 mL of water, and urination, whole-body PET/CT (discovery VCT^®^, GE Healthcare, Milwaukee, WI, USA) examination was performed. First, 64 slice spiral CT (Discovery VCT^®^, GE Healthcare, Milwaukee, WI, USA) scanning was performed, operating with tube voltage 110 kV, tube current 110 mA, and layer thickness 3.3 mm. Then, PET imaging was acquired in the three-dimensional acquisition mode, scanning the range from the upper to the middle thigh with 2 min per bed, for a total of 6–7 beds. PET data were reconstructed with the ordered subsets expectation maximization (OSEM) method. The standard reconstruction was performed with a 512 × 512 matrix and 3.3 mm slice thickness. CT data were used for attenuation correction. Finally, cross-sectional, sagittal, coronal CT, PET, and PET/CT fusion images were obtained by the GE AW4.6 workstation software (GE Healthcare, Milwaukee, WI, USA).

### 2.3. ^18^F-FDG PET/MR Scan

After PET/CT acquisition, all patients underwent a neck PET/MR (3.0 T, signa TOF-PET/MR, GE Healthcare, Milwaukee, WI, USA) scan around 120 min after ^18^F-FDG injection. Head and neck coil and field of view (FOV) were used for PET/MR scanning. The scanning sequence included axial fast spin echo T1 weighted imaging (FSE T1WI), axial, sagittal, and coronal FSE T2WI, and finally axial DWI. PET scanning used 3D acquisition, with about 15 min per bed. The scanning parameters were T1WI (turn angle = 142°, echo time [TE]/repeat time [TR] = 13.2/640 ms, bandwidth = 41.67 KHz, FOV = 24 cm × 24 cm, matrix = 256 × 192), T2WI (turn angle = 142°, echo time [TE]/repeat time [TR] = 161/3054 ms, bandwidth = 50 KHz, FOV = 22 cm × 22 cm, matrix = 288 × 288), DWI (echo time [TE]/repeat time [TR] = 191/2500 ms, bandwidth = 250 KHz, FOV = 24 cm × 24 cm, matrix = 96 × 96, b = 600). The time of flight (TOF) technique and OSEM algorithm were used to reconstruct the PET data with the following parameters: FOV = 30 cm × 30 cm, matrix = 192 × 192, filter cutoff = 3.0 mm, subset = 28, iteration = 3. PET attenuation correction was based on atlas MR attenuation correction combined with the Dixon water–fat separation method.

### 2.4. Image Interpretation and Analysis

All acquired images were analyzed by the AW workstation. Two nuclear medicine physicians (X.L., with 24 years of experience and F.L. with 20 years of experience in radiology and 5 years of experience in nuclear oncology) visually interpreted PET/CT and PET/MR images and collected the following information: (1)lesion counts detected on PET/CT and PET/MR, respectively;(2)lesion conspicuity scoring according to [26] (assessed independently): 1 point for no detection; 2 points for detected suspicious morphological correlation; 3 points for clear morphological correlation;(3)lesion diagnosis (any difference of opinion resolved by consensus): benign or malignant.

Two nuclear medicine physicians (Y.S. and F.L.) delineated the region of interest (ROI) along the edge of the focus segmented on T1WI/T2WI and CT according to the aforementioned detected lesions and collected the following dataset respectively:(1)lesion diameters: long and short diameters as referred in RECIST 1.1 [27];(2)standardized uptake value (SUV) for lesions calculated automatically by the workstation: SUVmax and SUVmean.

### 2.5. Local Recurrence and Metastatic Lymph Nodes

The gold standard for locally recurrent DTC or nodal metastases was determined according to one of the following criteria: (1) histopathological diagnosis of recurrence or metastasis; (2) clinical serum Tg and/or TgAb levels increased continuously during the follow-up of more than 6 months and imaging (neck ultrasound, CT, and/or MR) revealing following malignant features simultaneously.

Ultrasound criteria: nodules or lymph nodes were considered malignant if the short axis diameter was ≥10 mm in levels I–II or ≥7 mm in levels III–VI, the volume increased more than 50%, or the diameter increased more than 20% or 2 mm. In addition, other signs of malignancy, including spherical or long-to-short axis ratio < 2, absence of an echogenic hilum, microcalcification and cystic changes, could classify the node as malignant, regardless of the size of the lymph node [28,29,30,31].

CT or MR criteria: recurrence was determined by size and abnormal density/signal with irregular edges or blurred boundaries on CT/MR; for nodules or lymph nodes, they were perceived as malignant if the maximum axial diameter was ≥8 mm in the retropharyngeal space, ≥15 mm in levels I–II or ≥10 mm in levels III–VII, or the volume increased more than 50%, or the diameter increased more than 20% or 2 mm [32,33]. In addition, other signs of malignancy, including central necrosis, contrast enhancement, intralesional calcifications and cystic changes, could classify the nodule as malignant, irrespective of nodal size [34].

### 2.6. Statistical Analyses

SPSS 26.0 (IBM, Armonk, New York, NY, USA) and GraphPad Prism 9.0 (GraphPad Software Inc., San Diego, CA, USA) were used for statistical analysis and figure production, respectively. For all variables, the Kolmogorov–Smirnov normal distribution test was performed first. Continuous variables with a normal distribution were expressed as mean ± SD. Continuous variables that were not normally distributed were expressed as median and interquartile intervals. Categorical variables were expressed as numbers and percentages. The Kappa consistency analysis was used to assess the two physicians’ subjective score, and the Wilcoxon signed-ranks test was used to evaluate the image conspicuity difference. The sensitivities, specificities, positive predictive values (PPVs), negative predictive values (NPVs), and accuracies of PET/MR, MR, and PET/CT for diagnosis were determined in accordance with the gold standard. According to the R × C chi-square test and the fisher exact probability method, the difference in diagnostic performance of PET/MR, MR, PET/CT was tested. The Wilcoxon signed-rank test was used to test the difference of PET/MR and PET/CT parameters (SUVmax, SUVmean, and lymph node diameter) for the same lymph nodes subgroup. The independent-samples Mann–Whitney U test was used to analyze the differences of these parameters between malignant and benign lymph nodes. Spearman’s correlation method was used to analyze the correlation between PET/MR and PET/CT parameters and calculate the coefficient of determination (r^2^). Bland–Altman analysis was used to compare the two techniques.

## 3. Results

A total of 37 patients were retrospectively reviewed. Figure 1 shows the case screening. The patients (12 males and 25 females) had a mean age of 38.95 ± 12.03 years (range, 12–68 years). The time interval between PET examination and the first surgery ranged from 4 months to >5 years. In total, 25 patients had elevated non-stimulated Tg levels along with negative ^131^I-WBS, their median serum Tg was 11.01 ng/mL (range, 3.04– > 500 ng/mL), with thyrotropin levels all < 0 μIU/mL. The TgAb level of five patients rose to 2302.48 ± 1470.59 IU/mL (range 649.4– > 4000 IU/mL). One patient had an increase in both Tg and TgAb. After the PET/MR and PET/CT examinations, seven patients completed reoperation and histopathological examinations, and two patients underwent FNAB. The period of follow-up was between 8 and 24 months. Table 1 presents the general characteristics of the enrolled patients.

### 3.1. Patient-Based Analysis

Among the 37 patients, no malignant signs were found in 10 patients, while 24 patients had lymph node metastasis, and 3 patients presented coexisted recurrence with lymph node metastases. PET/MR correctly determined the disease status of 36 patients (97.3%), while PET/CT identified 31 cases (83.8%). Of the three coexisting cases, two were correctly identified by PET/MR and one was correctly identified by PET/CT. A recurrent focus was missed by both modalities. Of 10 negative patients, 3 presented pulmonary metastases on whole-body PET/CT, 4 were pathologically confirmed as negative, and the remaining 3 patients were found to have decreased Tg during follow-up. Figure 2 displays a patient with recurrence and lymph node metastasis, where PET/MR successfully identified the recurrent lesion in the thyroid region, but PET/CT missed it. Lymph node metastases in these patients were all successfully identified by both modalities. In the 24 patients with lymph node metastases detected, all were correctly diagnosed by PET/MR, and only 20 patients were recognized by PET/CT. Figure 3 shows a DTC patient with lymph node metastases, which were identified by PET/MR but missed by PET/CT.

### 3.2. Lesion-Based Analysis

Finally, a total of 130 lesions (9 in the original thyroid area, 121 in the cervical lymph node) were examined, including 3 malignant and 6 benign thyroid nodules, as well as 74 malignant and 47 benign cervical lymph nodes. Of all lesions, 44 (33.8%) were analyzed by histopathology, including 32 malignant lesions (1 local recurrence and 31 metastatic lymph nodes) and 12 benign lesions (2 benign thyroid nodules and 10 benign lymph nodes). Other lesions were confirmed by strict follow-up in the aforementioned manner. Among 130 lesions, PET/MR detected 119 (91.5%), but missed 1 recurrent thyroid cancer and 10 metastatic lymph nodes; PET/CT detected 105 (80.8%), but missed 2 recurrent thyroid cancers, 21 metastatic lymph nodes, 1 benign thyroid nodule, and 1 benign lymph node. For 74 lymph node metastases, the detection numbers of PET/MR and PET/CT were, respectively, 64 (86.5%) and 53 (71.6%). Table 2 shows the number of thyroid nodules and lymph nodes detected by ^18^F-FDG PET/MR and PET/CT.

The diagnostic performance of two PET modalities (PET/MR and PET/CT) and MR (alone) is displayed in the Table 3 and Figure 4. The diagnostic sensitivities of PET/MR, MR, and PET/CT were significantly different (80.5%, 77.9%, and 61.0%, *p* = 0.012). The paired comparison of PET/MR, MR, and PET/CT showed significant differences of sensitivity between PET/CT and the other two techniques (*p* < 0.001 and *p* = 0.007 respectively). The specificities of the three modalities (84.9%, 83.0%, and 81.1%, *p* = 0.875) showed no significant difference, also in paired comparisons. For metastatic lymph nodes, the sensitivities of PET/MR, MR, and PET/CT were 81.1%, 78.4%, and 62.2% (*p* = 0.018). Paired comparisons also showed significant differences between PET/CT and the other two techniques (*p* = 0.001 and *p* < 0.001 respectively). PPVs, NPVs, and accuracies in diagnosing all lesions (recurrent and malignant cervical lymph nodes) were generally consistent in tendencies among the three imaging modalities, among which PET/MR yielded the optimal diagnostic performance. Both imaging modalities yielded false-positive results, either for suspicious thyroid nodules or lymph nodes. Based on the available pathological results, misdiagnosed lesions were mainly fibrofatty tissue.

The image clarity scores for each region of the lesion are shown in Table 4. For all included lesions, the PET/MR scores were significantly higher than those of PET/CT (2.74 ± 0.60 vs. 1.9 ± 0.50, *p* < 0.001). The largest score was in level II (2.92 ± 0.33), which almost exceeded PET/CT (1.91 ± 0.45) by 1 point. For malignant lesions, there was no difference between PET/MR and PET/CT in the assessment of suspected thyroid nodules, while there were significant differences in the assessment of metastatic lymph nodes of all levels. The largest differences between PET/MR and PET/CT occurred in level II (2.83 ± 0.48 vs. 1.79 ± 0.58) and level V (2.73 ± 0.67 vs. 1.85 ± 0.46), as presented in Figure 3. For benign lesions, PET/MR had better presentation of thyroid nodules and lymph nodes in other regions, especially in levels II (3.00 ± 0.00 versus 2.02 ± 0.25) and IV (3.00 ± 0.00 versus 2.14 ± 0.36). Image conspicuity agreement was excellent (κ = 1 in thyroid nodules, κ = 0.981 in lymph nodes, *p* < 0.001) between the two physicians. 

Table 5 lists the SUVmax, SUVmean, and lymph node diameters in the maximum cross-sectional area of lymph nodes measured by PET/MR and PET/CT. The SUVmax of malignant lymph nodes was significantly higher than that of benign lymph nodes on PET/MR (median 2.6 vs. 2.2, *p* = 0.004) and PET/CT (median 2.0 vs. 1.8, *p* = 0.006). However, SUVmax showed an overlap in benign and malignant lymph nodes (Appendix A). Figure 5 presents an isolated malignant lymph node with intense ^18^F-FDG uptake on PET/MR. The SUVmean, long diameter, and short diameter indicated no significant differences between benign and malignant nodes on both modalities (Appendix A). For all included lymph nodes, SUVmax, SUVmean, and lymph node diameters of PET/MR were higher than those measured on PET/CT by 17.4%, 22.2%, 11%, and 18% (all *p* ≤ 0.001), respectively. The Bland–Altman analysis showed great consistency among modalities between the SUVmax, SUVmean, and diameters (Appendix A). Additionally, there were correlations between the parameters detected on PET/MR and PET/CT (all *p* < 0.001, Appendix A). The correlation coefficient of SUVmax for all lymph nodes was 0.661 (Appendix A).

## 4. Discussion

In this study, the head-to-head comparison of PET/MR with PET/CT in DTC patients after comprehensive treatment revealed that PET/MR was more accurate in determining recurrent/residual and metastatic lesions, both from a patient-based and a lesion-based perspective. PET/MR provided superior conspicuity, as shown in clarity score evaluation, making it more useful for the identification of lesions. Both malignant and benign lesions had a significantly higher SUVmax, making it essential to incorporate the morphological content provided by MR to strengthen diagnosis efficacy. Therefore, additional neck PET/MR may be recommended for the detection of recurrence/lymph nodes metastases in DTC patients after comprehensive treatment, as it provides clearer images, more accurate identification, and more precise resection scope to avoid over-resection.

PET/MR can identify recurrent lesions and metastatic lymph nodes more accurately. According to patient analysis, PET/MR detected 26 of 27 patients with tumor burden, while PET/CT only detected 21. Based on lesion analysis, PET/MR combined with the advantages of MR had higher detection rates (91.5% vs. 80.8%), image conspicuity (especially in level II), and diagnostic efficacy than PET/CT. It was found that PET/MR showed higher diagnostic efficacy, and the MR part of PET/MR showed different positive results compared to ^18^F-FDG PET/CT, which suggested that the combination of the two modalities had a synergistic effect. This is consistent with published research [16,35]. However, there are still false positives for both imaging modalities, which are mainly caused by local postoperative hyperplasia. Based on the available pathological diagnosis in this study, suspicious areas depicted on PET/MR or PET/CT were fibrofatty tissue closely associated with postoperative local hyperplasia. In addition, high physiological ^18^F-FDG uptake can be observed in Waldeyer’s ring, active muscles (vocal cord movement, swallowing), salivary glands, and brown fat [36].

MR plays an important role in the detection of complex thyroid lesions and metastatic lymph nodes [18,19,20] not only because MR has superior soft tissue contrast and additional imaging techniques, but also because multi-position, multi-parameter, and multi-sequence images can provide several benefits. These benefits include clear lesion contour and location, signal alteration and enhancement, diffusion restriction, and accurate assessment of lesion invasion of surrounding tissue structures (e.g., envelope, cartilage) [37]. On conventional MR, malignant thyroid nodules exhibit irregular margins and blurred boundaries with cystic changes, diffuse enlargement of the thyroid gland, and peripheral and distant invasion. They usually reveal low to intermediate T1- and high T2-weighted signal intensities, or occasionally high T1- and T2-weighted signal intensities with heterogeneous enhancement on enhanced T1WI [38]. On DWI, recurrence or metastases present strong contrast with the dark background signal from fat deposited around, based on metabolic or physiological changes [39,40]. Additionally, one should note that the clear volumetric information provided by MRI may be meaningful for determining the treatment dose [41]. In regard to image conspicuity, PET/MR had obvious advantages over PET/CT in detecting and displaying thyroid nodules as well as suspicious lymph nodes regardless of the region of interest. Compared with PET/CT, high resolution and multiple sequences such as DWI and ADC of MR have made contributions to the imaging of lesions, especially in level II with complex anatomy and high incidence of nonspecific lymph nodes [42]. Of note, the updated NCCN Guidelines of Thyroid Carcinoma (Version 3.2020) recommend that if iodine-131 imaging is negative and stimulated Tg > 2–5 ng/mL, additional non-radioiodine imaging modalities should be considered, including central and lateral neck compartments ultrasound, neck CT with contrast, chest CT with contrast, and PET/CT. However, enhanced CT is not routinely performed in patients with highly suspected recurrence or metastases in our department, in order to avoid the influence of the iodine contrast agent on the subsequent ^131^I treatment.

^18^F-FDG PET/CT is mainly used for postoperative surveillance of poorly differentiated thyroid cancer [43]. The uptake ability of ^18^F-FDG by thyroid cancer depends on the tumor differentiation degree [44,45]. The lower the differentiation of the lesion, the less it takes up ^131^I, but the more it takes up ^18^F-FDG, so positive Tg and negative ^131^I-WBS lesions are prone to be identified. This study also verified that SUVmax associated with ^18^F-FDG uptake was meaningful for the characterization of cervical lymph nodes. For PET alone, modern PET/MR can provide a superior SUVs than PET/CT, with time-of-flight (TOF) technology based on a new generation of crystals and the latest photovoltaic conversion technology.

There are contradictory results on the value of PET/CT and additional MR according to previous research. On the one hand, Hempel et al. concluded that the combination of PET/CT with MR was not suitable for routine clinical application [16,46]. Loeffelbein et al. reported that there was no significant difference between PET, MRI, and PET + MR in identifying recurrences in the neck [47]. On the other hand, earlier research argued that PET/CT and additional MR can provide additional diagnostic information [48,49]. PET/MR allows accurate, temporally and spatially unified multiparametric imaging of PET and MR. It can provide additional information and further improve the diagnostic efficacy than non-simultaneous PET and MR [50]. Previous studies have reported that PET/MR did not outperform PET/CT or even slightly underperform PET/CT in the detection of lung metastases and bone metastases [26,41]. Our results reveal that PET/MR is comparable to PET/CT in terms of diagnostic specificity. In addition, for the same lesion, SUVmax, SUVmean, and diameters measured on PET/MR and PET/CT were consistent and had significant correlation. Although there were significant differences between the measured parameters of PET/MR and PET/CT, this may be related to different tracer elimination, attenuation correction, and associated scatter correction.

There are several limitations in this study. First, the number of patients and lesions involved was limited. Most lesions were determined by a follow-up combination of Tg levels and imaging (ultrasound, CT, and/or MR) instead of surgical biopsy or FNAB; therefore, the absence of histopathological confirmation of most lesions may potentially affect the final diagnostic accuracy. Second, quantitative comparisons of parameters were limited by different imaging techniques, different attenuation correction and associated scatter correction methods, as well as different tracer elimination caused by the single-injection double-examination protocol. In addition, the study was retrospective and only included patients with DTC who underwent both PET/CT and PET/MR. Multicenter clinical trials that prospectively include patients are necessary to assess the validity of our results. Furthermore, cervical lymph nodes are often inflammatory, which may lead to nonspecific ^18^F-FDG uptake. Therefore, the exploitation of specific imaging agents should be the orientation of future research. In this study, although ^18^F-FDG uptake differed between malignant and benign disease, there was also overlap in both SUV values, which may have an impact on the accuracy of the results. Third and lastly, PET/MR examinations are expensive, and the clinical application of this study should also consider the overall benefit to patients in relation to financial expenditures. Further analysis of the economic benefits is expected.

## 5. Conclusions

This study compared ^18^F-FDG PET/CT and additional neck PET/MR scans head to head to analyze the detection ability and diagnostic efficacy of recurrent/residual DTC lesions and cervical lymph node metastases, as well as the characteristics of each parameter. The results showed that ^18^F-FDG PET/MR was more accurate in diagnosing recurrent and metastatic lesions, both from a patient-based and a lesion-based perspective. PET/MR detected more lesions and presented better image conspicuity and diagnostic performance due to the high soft tissue resolution and multi-sequence imaging available with MRI. The addition of neck PET/MR after whole-body PET/CT may be recommended to provide more precise diagnostic information and scope of surgical resection without additional ionizing radiation. Further prospective studies with expanded samples and economic benefit analysis are needed to support the conclusions.

## Figures and Tables

**Figure 1 cancers-13-03436-f001:**
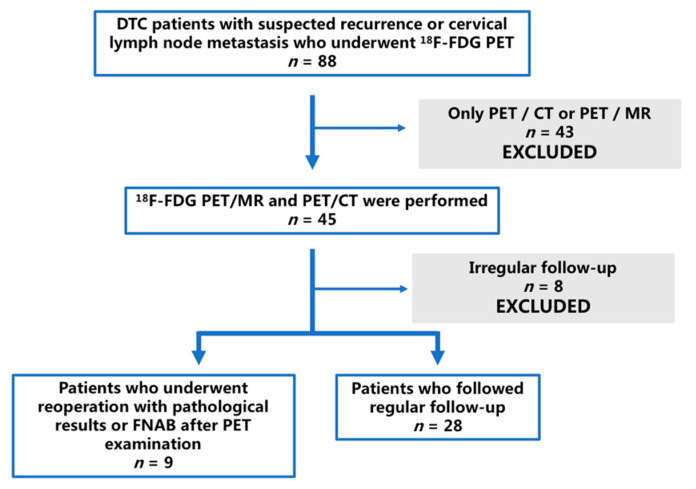
Flow chart of patients who were referred for assessment.

**Figure 2 cancers-13-03436-f002:**
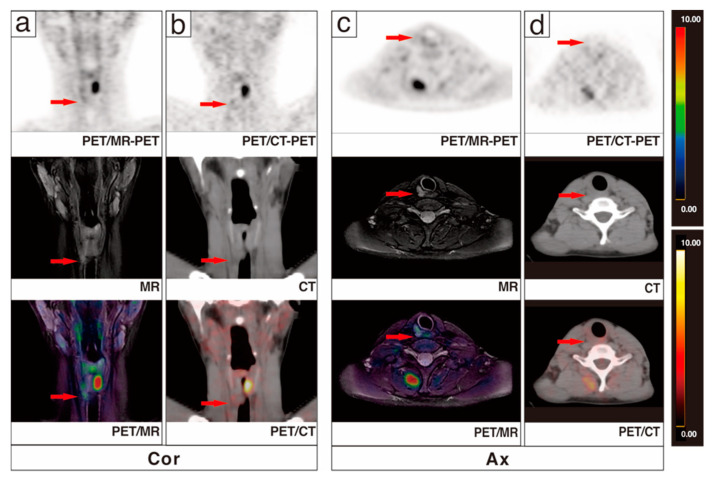
A 17-year-old post-operative DTC patient underwent ^131^I treatment three times with persistently elevated Tg up to 371.0 ug/L. ^131^I-WBS showed scattered iodine uptake foci in the right paratracheal area and lung lobes. ^18^F-FDG PET/CT was performed for whole-body systemic evaluation. ^18^F-FDG PET/CT showed no significant abnormalities in the thyroid region ((**b**,**d**), red arrows); additional PET/MR of the neck showed a long T2 signal nodule in the right thyroid region with a mild metabolic increase SUVmax 1.7 ((**a**,**c**), red arrows). In combination with medical history and ^131^I examination, a residual/recurrent thyroid cancer was diagnosed. The patient was subsequently reoperated, and residual/recurrent thyroid cancer was confirmed by histological pathology.

**Figure 3 cancers-13-03436-f003:**
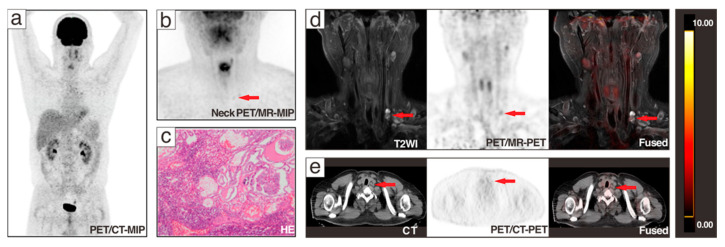
A 36-year-old patient with Tg 39.1 μg/L and negative ^131^I-WBS after comprehensive treatment was admitted to ^18^F-FDG PET/CT and underwent additional neck PET/MR. PET/MR clearly presented the morphological and metabolic changes of lymph nodes in the VB region ((**b**,**d**), red arrows). PET/CT showed no significant changes in the corresponding lesions ((**a**,**e**), red arrows). After reoperation, 1 lymph node in the left VB, 8 lymph nodes in the left level IIA, 7 lymph nodes in the left level III, and 10 lymph nodes in the left level IV were taken for pathomorphological examination, and only the lymph node in the VB was pathologically diagnosed to be a metastatic lymph node (**c**).

**Figure 4 cancers-13-03436-f004:**
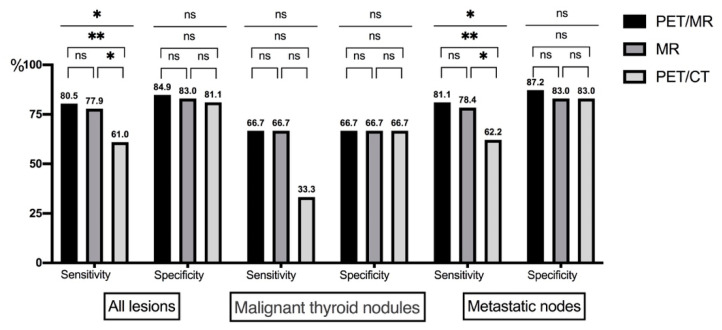
Diagnostic performance of two modalities (ns represents no significance; * represents the *p*-value < 0.05; ** represents the *p*-value < 0.01).

**Figure 5 cancers-13-03436-f005:**
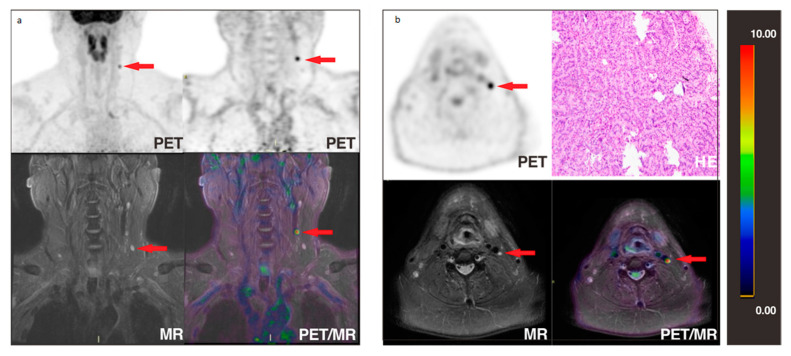
A 51-year-old DTC patient after comprehensive treatment presented with an unexplained increase in Tg (2.88 μg/L). PET/MR depicted an increased uptake of a small lymph node (long diameter, 0.7 cm) in the left level III, with SUVmax 5.1, without other apparently abnormal lymph nodes ((**a**,**b**), red arrows). 30 suspicious lymph nodes were removed by re-operation, and notably, only the lymph node identified on PET/MR was malignant.

**Table 1 cancers-13-03436-t001:** Patients’ characteristics.

Characteristics		Number
Age	38.95 ± 12.03 years (range, 12–68 years)	
	<55 years	33 (89.2%)
	>55 years	4 (10.8%)
Sex	Male	12 (32.4%)
	Female	25 (67.6%)
Histologic type	PTC ^1^	29 (78.4%)
	PTMC ^2^	4 (10.8%)
	PTC ^1^ + PTMC ^2^	3 (8.1%)
	FTC ^3^	1 (2.7%)
Stage of disease	I	32 (86.5%)
	II	2 (5.4%)
	IV	3 (8.1%)
Clinical indication for PET Imaging	Positive Tg ^4^ and negative ^131^I-WBS ^5^	25 (67.6%)
	Rising TgAb ^6^ after radioactive iodine ablation	4 (10.8%)
	Suspected metastasis detection	8 (21.6%)
Gold standard source	Pathology after reoperation	7 (18.9%)
	Fine-needle aspiration	2 (5.4%)
	Regular follow-up	28 (75.7%)

^1^ PTC papillary thyroid carcinoma, ^2^ PTMC papillary thyroid microcarcinoma, ^3^ FTC follicular thyroid carcinoma, ^4^ Tg thyroglobulin, ^5^
^131^I-WBS ^131^Iodine whole body scan, ^6^ TgAb anti-thyroglobulin antibodies.

**Table 2 cancers-13-03436-t002:** Number of thyroid nodules and lymph nodes detected by ^18^F-FDG PET/MR and PET/CT.

		Golden Standard	PET/MR	PET/CT
Malignant	Thyroid nodules	3	2	1
	Lymph nodes	74	64	53
Benign	Thyroid nodules	6	6	5
	Lymph nodes	47	47	46
Total		130	119	105

**Table 3 cancers-13-03436-t003:** Diagnostic performance of two modalities.

	All Lesions			Malignant Thyroid Nodules			Metastatic Nodes		
	PET/MR	MR	PET/CT	PET/MR	MR	PET/CT	PET/MR	MR	PET/CT
True-positive	62	60	47	2	2	1	60	58	46
True-negative	45	44	43	4	4	4	41	40	39
False-positive	8	9	10	2	2	2	6	7	8
False-negative	15	17	30	1	1	2	14	16	28
Sensitivity (%)	80.5%	77.9%	61.0%	66.7%	66.7%	33.3%	81.1%	78.4%	62.2%
Specificity (%)	84.9%	83.0%	81.1%	66.7%	66.7%	66.7%	87.2%	85.1%	83.0%
PPV (%)	88.6%	87.0%	82.5%	50.0%	50.0%	33.3%	90.9%	89.2%	85.2%
NPV (%)	75.0%	72.1%	58.9%	80.0%	80.0%	66.7%	74.5%	71.4%	58.2%
Accuracy (%)	82.3%	80.0%	69.2%	66.7%	66.7%	55.6%	83.5%	81.0%	70.2%

**Table 4 cancers-13-03436-t004:** Conspicuity score of recurrent or metastatic lesions in relation to location (scoring criteria referring to [26]).

		All Lesions				Malignant				Benign			
		No.	PET/MR	^18^F-PET/CT	*p*	No.	PET/MR	PET/CT	*p*	No.	PET/MR	^1^PET/CT	*p*
Thyroid area		9	2.56 ± 0.73	2.00 ± 0.707	0.027	3	2.00 ± 0.894	2.00 ± 0.894	1	6	2.83 ± 0.39	2.00 ± 0.60	0.008
Lymph nodes	II	53 (43.8%)	2.92 ± 0.33	1.91 ± 0.45	<0.001	24 (45.3%)	2.83 ± 0.48	1.79 ± 0.58	<0.001	29 (54.7%)	3.00 ± 0.00	2.02 ± 0.25	<0.001
	III	16 (13.2%)	2.38 ± 0.79	1.84 ± 0.52	<0.001	13 (81.3%)	2.31 ± 0.84	1.81 ± 0.57	0.003	3 (18.8%)	2.67 ± 0.52	2.00 ± 0.00	0.46
	IV	18 (14.9%)	2.67 ± 0.68	1.97 ± 0.56	<0.001	11 (61.1%)	2.45 ± 0.82	1.82 ± 0.60	0.035	7 (38.9%)	3.00 ± 0.00	2.14 ± 0.36	0.001
	V	19 (15.7%)	2.74 ± 0.64	1.92 ± 0.54	<0.001	13 (68.4%)	2.73 ± 0.67	1.85 ± 0.46	<0.001	6 (31.6%)	2.75 ± 0.62	2.08 ± 0.67	0.11
	VI	15 (12.4%)	2.50 ± 0.76	1.73 ± 0.45	0.002	13 (86.7%)	2.54 ± 0.74	1.82 ± 0.55	0.002	2 (13.3%)	3.00 ± 0.00	2.50 ± 0.71	<0.001
	Total	121	2.74 ± 0.60	1.9 ± 0.50	<0.001	74	2.61 ± 0.72	1.8 ± 0.57	<0.001	47	2.96 ± 0.20	2.04 ± 0.29	<0.001
		Kappa = 0.981, *p* < 0.001											

**Table 5 cancers-13-03436-t005:** SUVs, diameters of lymph nodes detected by both PET/MR and PET/CT.

		PET/MR	PET/CT
All detected lymph nodes *n* = 99	SUVmax	2.3 (2.0–2.8) ^1^	1.9 (1.5–2.2)
	SUVmean	1.8 (1.5–2.2)	1.4 (1.1–1.8)
	Long axis	9.4 (7.2–11.7)	7.3 (5.3–9.5)
	Short axis	5.0 (4.0–6.2)	4.1 (3.0–5.3)
Malignant *n* = 54	SUVmax	2.6 (2.0–3.5)	2.0 (1.6–2.3)
	SUVmean	1.9 (1.5–2.5)	1.4 (1.2–1.9)
	Long axis	9.7 (7.3–11.4)	7.3 (5.0–9.4)
	Short axis	4.7 (3.8–6.9)	3.7 (3.0–5.0)
Benign *n* = 45	SUVmax	2.2 (1.9–2.6)	1.8 (1.5–2.0)
	SUVmean	1.7 (1.5–2.0)	1.4 (1.0–1.8)
	Long axis	8.9 (7.0–12.7)	7.3 (5.5–9.6)
	Short axis	5.3 (4.5–6.1)	4.4 (3.3–5.4)

^1^ The data are uniformly expressed as median (25th–75th %).

## Data Availability

The data presented in this study are available in this article (and Appendix A).

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
