# Peer review of "Head-to-Head Comparison of Neck 18F-FDG PET/MR and PET/CT in the Diagnosis of Differentiated Thyroid Carcinoma Patients after Comprehensive Treatment"

_cancers, 2021, doi:10.3390/cancers13143436_

Round 1
Reviewer 1 Report
The authors evaluated the clinical value of 18F-FDG PET/MR in head-to-head comparison of PET/CT in recurrent and metastatic cervical lymph nodes of DTC patients. As as result, compared with 18F-FDG PET/CT, PET/MR was more accurate in determining recurrent and metastatic lesions, both from a patient-based and a lesion-based perspective although number of patients is still limited.
In the previous study (Vrachimis A, et al. Eur J Nucl Med Mol Imaging. 2016), it was reported that FDG-PET/MRI was inferior to low-dose FDG-PET/CT for the assessment of lung metastasis, and FDG-PET/MRI was equal to contrast-enhanced neck FDG-PET/CT for assessment of cervical status in patients with thyroid cancer.
This study showed the superiority of PET/MR compared to PET/CT, focusing on the neck regions.
I have some comments.
1) In the CT or MR criteria for the recurrence, increase in size during follow-up should be added to the size criteria like ultrasound criteria,
except for the lymph nodes which have apparent signs of malignancy.
2) Please explain what kind of findings in which sequence of MR is useful to detect recurrence compared to PET/CT in detail.
3) What is the reason of false positive cases by PET/MR and PET/CT.
Author Response
Response to Reviewer #1:
Reviewer #1: The authors evaluated the clinical value of 18F-FDG PET/MR in head-to-head comparison of PET/CT in recurrent and metastatic cervical lymph nodes of DTC patients. As a result, compared with 18F-FDG PET/CT, PET/MR was more accurate in determining recurrent and metastatic lesions, both from a patient-based and a lesion-based perspective although number of patients is still limited.
In the previous study (Vrachimis A, et al. Eur J Nucl Med Mol Imaging. 2016), it was reported that FDG-PET/MRI was inferior to low-dose FDG-PET/CT for the assessment of lung metastasis, and FDG-PET/MRI was equal to contrast-enhanced neck FDG-PET/CT for assessment of cervical status in patients with thyroid cancer.
This study showed the superiority of PET/MR compared to PET/CT, focusing on the neck regions.
I have some comments.
Response: Thanks for your positive comment.
We have carefully revised our manuscript to improve our paper quality according to your suggestions.
Comment 1: In the CT or MR criteria for the recurrence, increase in size during follow-up should be added to the size criteria like ultrasound criteria, except for the lymph nodes which have apparent signs of malignancy.
Response: Thanks for your helpful suggestion.
We added the description of internal growth in the CT or MR criteria and recharacterized all follow-up lesions according to the updated criteria. The modified criteria were shown as follows:
For nodules or lymph nodes, they were perceived as malignant if the maximum axial diameter was ≥8 mm in the retropharyngeal space, ≥15 mm in levels I-II or ≥10 mm in levels III–VII, or the volume increased more than 50%, or the diameter increased more than 20% or 2 mm.
Comment 2: Please explain what kind of findings in which sequence of MR is useful to detect recurrence compared to PET/CT in detail.
Response: Thank you for your very important suggestion.
We explained the specific findings on different sequences of MR which may be useful for diagnosis of the recurrence in the Discussion section, and shown as follows:
On conventional MR, malignant thyroid nodules exhibit irregular margins and blurred boundaries with cystic changes, diffuse enlargement of the thyroid gland, and peripheral and distant invasion. They usually reveal low to intermediate T1- and high T2-weighted signal intensities or occasionally high T1- and T2-weighted signal intensities, with heterogeneous enhancement on enhanced T1WI [#1]. In addition, on diffusion-weighted imaging (DWI), recurrence or metastases present strong contrast with the dark background signal from fat deposited around, based on metabolic or physiological changes [#2, #3].
[#1] Som, P.M.; Brandwein, M.; Lidov, M.; Lawson, W.; Biller, H.F. The varied presentations of papillary thyroid carcinoma cervical nodal disease: CT and MR findings. Ajnr American Journal of Neuroradiology 1994; 15, 1123.
[#2] Park, S.-H.; Hahm, M.H.; Bae, B.K.; Chong, G.O.; Jeong, S.Y.; Na, S.; Jeong, S.; Kim, J.-C. Magnetic resonance imaging features of tumor and lymph node to predict clinical outcome in node-positive cervical cancer: a retrospective analysis. Radiation Oncology 2020; 15, 86.
[#3] Mao, Y.; Hedgire, S.; Harisinghani, M. Radiologic Assessment of Lymph Nodes in Oncologic Patients. Current Radiology Reports 2014; 2, 36.
Comment 3: What is the reason of false positive cases by PET/MR and PET/CT.
Response: Thank you for raising up this important question.
False positives are mainly caused by local postoperative hyperplasia and pharyngeal inflammation. Based on the available pathological diagnosis in this study, it indicated that suspicious areas depicted on PET/MR or PET/CT were actually fibrofatty tissue closely associated with postoperative local hyperplasia. In addition, high physiological 18F-FDG uptake can be observed in Waldeyer's ring, active muscles (vocal cord movement, swallowing) salivary glands, and brown fat [#4].
We have added descriptions of the false-positive reasons in the Results and Discussion sections.
[#4] Abouzied, M.M.; Crawford, E.S.; Nabi, H.A. 18F-FDG imaging: pitfalls and artifacts. J Nucl Med Technol 2005; 33, 145-155; quiz 162-143.
Reviewer 2 Report
The matter that PET-MR would have some Benefits in the cervical area is known.
The Question is, what would be additional value of MRT to PET-CT, rather than PET-MR?
Additional MRT only or fusion of MRT with PET Images would add the same Information to the clinician.
This must be discussed by the authors. The comment "that PET-CT is not yet replaced by PET-CT" does not reflect the reality and should be removed. PET-MR will be a good infrastructure in the University Hospitals, but PET-CT will remain the Gold Standard in the Routine Hospitals.
Author Response
Response to Reviewer #2:
Comment 1: The matter that PET-MR would have some Benefits in the cervical area is known.
The Question is, what would be additional value of MRI to PET-CT, rather than PET-MR?
Additional MRI only or fusion of MRI with PET Images would add the same Information to the clinician. This must be discussed by the authors.
Response: We appreciate your comments and suggestions.
It is important to know the clinical importance and the different diagnosis efficacy between PET/MR and PET/CT with MR. Theoretically, addition of MR alone after PET/CT may be same as the addition of PET/MR. However, there are contradictory results on the value of PET/CT and additional MR according to the previous researches. On the one hand, Hempel et al. concluded that the combination of PET/CT with MR was not suitable for routine clinical application. Although PET/CT and MR improved the sensitivity and negative predictive value (NPV), it is noticed that the specificity, positive predictive value (PPV) and diagnostic accuracy were significantly reduced [#1, #2]. Loeffelbein et al. identified that there was no significant difference between PET, MRI, and PET+MRI in identifying recurrences in the neck [#3]. On the other hand, some researches argued that fused images of PET/CT and additional MR can provide additional diagnostic information [#4, #5].
Hybrid PET/MR allows accurate, temporally and spatially unified multiparametric imaging of PET and MR. It can provide additional more information and further improve the diagnostic efficacy than non-simultaneous PET and MR [#6]. Furthermore, imaging machines in the Nuclear Medicine center are usually two-in-one like PET/CT and PET/MR, where two acquisitions in one scan are in principle more efficient than one acquisition in one scan. However, the drawback of PET/MR is expensive in terms of machines and examination cost cannot be ignored.
Our conclusions are derived from the methodology of the study, where additional neck PET/MR scans were performed after whole-body PET/CT. We did not conduct a separate comparison of differences in additional MR or PET/MR after PET/CT. We agree with the reviewer that relevant further researches are needed to provide more accurate imaging protocols for the follow-up of thyroid cancer patients, and head-to-head comparation of PET/CT+ neck PET/MR and PET/CT + MR. A relevant discussion has been added to the discussion.
[#1] Hempel, J.M.; Kloeckner, R.; Krick, S.; Pinto Dos Santos, D.; Schadmand-Fischer, S.; Boeßert, P.; Bisdas, S.; Weber, M.M.; Fottner, C.; Musholt, T.J.; et al. Impact of combined FDG-PET/CT and MRI on the detection of local recurrence and nodal metastases in thyroid cancer. Cancer Imaging 2016; 16, 37.
[#2] Subesinghe, M.; Scarsbrook, A.F.; Sourbron, S.; Wilson, D.J.; McDermott, G.; Speight, R.; Roberts, N.; Carey, B.; Forrester, R.; Gopal, S.V.; et al. Alterations in anatomic and functional imaging parameters with repeated FDG PET-CT and MRI during radiotherapy for head and neck cancer: a pilot study. BMC Cancer 2015; 15, 137.
[#3] Loeffelbein, D.J.; Souvatzoglou, M.; Wankerl, V.; Dinges, J.; Ritschl, L.M.; Mücke, T.; Pickhard, A.; Eiber, M.; Schwaiger, M.; Beer, A.J. Diagnostic value of retrospective PET-MRI fusion in head-and-neck cancer. BMC Cancer 2014; 14, 846.
[#4] Nakamoto, Y.; Tamai, K.; Saga, T.; Higashi, T.; Hara, T.; Suga, T.; Koyama, T.; Togashi, K. Clinical value of image fusion from MR and PET in patients with head and neck cancer. Mol Imaging Biol 2009; 11, 46-53.
[#5] Seiboth, L.; Van Nostrand, D.; Wartofsky, L.; Ousman, Y.; Jonklaas, J.; Butler, C.; Atkins, F.; Burman, K. Utility of PET/neck MRI digital fusion images in the management of recurrent or persistent thyroid cancer. Thyroid 2008; 18, 103-111.
[#6] Rosenkrantz, A.B.; Friedman, K.; Chandarana, H.; Melsaether, A.; Moy, L.; Ding, Y.S.; Jhaveri, K.; Beltran, L.; Jain, R. Current Status of Hybrid PET/MRI in Oncologic Imaging. AJR Am J Roentgenol 2016; 206, 162-172.
Comment 2: The comment "that PET-MR is not yet replaced by PET-CT" does not reflect the reality and should be removed. PET-MR will be a good infrastructure in the University Hospitals, but PET-CT will remain the Gold Standard in the Routine Hospitals.
Response: We appreciate your very helpful suggestions.
We totally agree with the reviewer that PET/MR will become a good infrastructure for university hospitals, but PET-CT will remain the gold standard for conventional hospitals. We have removed the improper statement to improve our paper quality according to the suggestion.
Reviewer 3 Report
In my opinion, the benefit of PET-MRI in malignant head and neck diseases is not a question of insufficient evidence of the modality itself, but of availability and health insurance coverage. A head-to-head comparison has to be very precise in patient selection and has to clearly specify and discuss the reasons why some findings might be better detected in one or the other modality. Furthermore, the golden standard must be a histopathological confirmation of the suspected recurrence or lymph node metastases.
In 27% (n=10) of the patients no malignant sign was found. Why did they undergo an imaging study? Did all of them had an elevated Tg-level? This should be claryfied in the result section. The only statement I found (table 1.) was that 67.6% had a positive thyroglobuline. Which of the 25 patients with elevated Tg had negative Imaging? Why would one go on with imaging studies in patients with TgAB positive Levels after RIT - if there is no further evidence for recurrence or LN-metastases? In which modality (presume: ultrasonography?) was a suspected metastasis detected?
Why was the tumor recurrence in one patient missed by both modalities and how was it proven to be a recurrance. Was it seen by ultrasound or due to follow up (with what: Tg, ultrasound, …) ?
Why did you not use a contrast enhanced CT? Comparing native CT with MRI is not valid, especially in lymphnode detection. Most of the patients (67.6%) had a negative 131-I scan, though there is no reason for not giving CT contrast media, except problems might gain with sequential PET-MRI. Please argue in the discussion section at least why you did not use CECT.
The method part does not go into exactly whether the examinations were carried out sequentially, which I assume. This means that a different time interval was introduced between FDG application and PET-CT and PET-MRI image recording. The pharmacokinetics shows, however, that the longer the uptake time, the better the contrast, since the bloodpool background is reduced. This phenomenon could result (together with the lower resolution of the old PET-CT system) in the somewhat better detection of PET-MRI as shown in fugure 2.
Another phenomenon is the higher SUV values aquired with PET-MRI compared to PET-CT. This might be the better intrinsic resolution and detector sensitivity from PET-MRI scanners or also the reconstruction algorithm used. There should be a phantom study comparing the calculated values if the scanners are in the same facility.
Furthermore, comparing an old system like the GE dicovery VCT PET scanner with a modern PET-MRI is equivalent to comparing apples and pears. The system has a BGO crystal and neither time-of-fleight nor count recovery reconstruction algorithms implemented. This limitation should at least have been mentioned in the discussion.
Round 2
Reviewer 1 Report
No additional comments.